# Non-obligate pairwise metabolite cross-feeding suggests ammensalic interactions between *Bacillus amyloliquefaciens* and *Aspergillus oryzae*

Digar Singh[1], Sang Hee Lee[1] & Choong Hwan Lee [ID] [1✉]

Bacterial-fungal metabolite trade-offs determine their ecological interactions. We designed a non-obligate pairwise metabolite cross-feeding (MCF) between *Bacillus amyloliquefaciens* and *Aspergillus oryzae*. Cross-feeding *Aspergillus* metabolites (MCF-1) affected higher growth and biofilm formation in *Bacillus*. LC-MS/MS-based multivariate analyses (MVA) showed variations in the endogenous metabolite profiles between the cross-fed and control *Bacillus*. We observed and validated that *Aspergillus*-derived oxylipins were rapidly depleted in *Bacillus* cultures concomitant with lowered secretion of cyclic lipopeptides (CLPs). Conversely, *Bacillus* extracts cross-fed to *Aspergillus* (MCF-2) diminished its mycelial growth and conidiation. Fungistatic effects of *Bacillus*-derived cyclic surfactins were temporally reduced following their hydrolytic linearization. MVA highlighted disparity between the cross-fed (MCF-2) and control *Aspergillus* cultures with marked variations in the oxylipin levels. We conclude that the pairwise MCF selectively benefitted *Bacillus* while suppressing *Aspergillus*, which suggests their ammensalic interaction. Widening this experimental pipeline across tailored communities may help model and simulate BFIs in more complex microbiomes.

[1] Department of Bioscience and Biotechnology, Konkuk University, 05029 Seoul, Korea. ✉email: chlee123@konkuk.ac.kr

Bacterial–fungal interactions (BFI) are widespread in nature and display a broad range of inter-kingdom ecological interactions, including both cooperative and competitive types. Besides physical adhesions, BFI are modulated by contactless chemical means through the release of nutritional compounds, secreted peptides, antibiotics, signaling molecules, and redox ions[1,2]. Considering the BFI, a major corollary is that metabolite cross-feeding (MCF) could reconcile the cooperative interactions contingent on compatible secretions under limiting conditions[3]. Although the evolutionary obligate cooperative interactions among microorganisms are theoretically and empirically interpreted, the non-obligate interactions remained largely unpredictable and poorly understood under non-limiting conditions. Factors like low culturability, high diversity, and lack of microbial model systems imitating the natural environment further complicate the laboratory models for MCF in BFIs[4,5].

BFIs play a vital role in driving major ecosystems that are indispensable to the environment, health, agriculture, and food[6,7]. BFIs contribute significantly to the biogeochemical cycles through nutrient and enzyme trade-offs that facilitate biomass recycling[8]. Considering BFIs in health, phenazines secreted from *Pseudomonas aeruginosa* impair biofilm formation and enhance conidiation in *Aspergillus fumigatus*, promoting its virulence[9,10]. BFI are important regulators of agriculture, where the plant growth-promoting rhizobacteria like *Streptomyces* and *Bacillus* selectively promote the ectomycorrhizal fungi while inhibiting the pathogen's growth, and hence modulate the tri-trophic interactions[11]. Secreted metabolites from bacteria may modulate the growth and metabolism of interacting fungi, resulting in effects such as reduced ethanol production in yeast and mitigation of antibacterial defense in *Aspergillus*[1,12]. BFIs are important determinants in various food fermentative bioprocesses employed toward the production of soy foods, dairy products, alcoholic beverages, and processed meats[13].

The importance of recurring *Bacillus* and *Aspergillus* interactions cannot be understated in food matrices. Both *Aspergillus oryzae* and *Bacillus amyloliquefaciens* are reported from the autochthonous mixed starters (*Nuruk*) used for soybean meju fermentation in traditional artisans[14,15]. Previously, we have shown that both *A. oryzae* and *B. amyloliquefaciens* are either used axenically or in-tandem toward the preparation of fermented soybean paste (doenjang meju) and koji[16–18]. We have reported the concomitant high abundance of *A. oryzae* and various *Bacillus* species including *B. subtilis*, *B. sonorensis*, *B. velenzensis*, *B. seohaeanensis* throughout the fermentative stages of doenjang meju[19]. Though the previously published data suggest the likely interactions between *A. oryzae* and *B. amyloliquefaciens*, the exclusive role of microbial metabolites in nutrient-rich food matrices is largely unexplored. Herein, we report the ecological functions of secretory secondary metabolites (SMs) in *B. amyloliquefaciens* KCCM 43033 and *A. oryzae* RIB 40 interactions beyond auxotrophies. Pairwise cross-feeding of the late-log phase metabolite extracts between *B. amyloliquefaciens* and *A. oryzae* selectively promoted bacterial growth and colonization while suppressing the fungi. Using the non-targeted metabolomics, we characterized the key metabolite classes which likely determined their ammensalic interactions following the MCF.

## Results

### MCF influences growth and developmental phenotypes in receiver species

*Metabolites from* A. oryzae *promote growth and biofilm formation in* B. amyloliquefaciens. Late-log phase *A. oryzae* culture extracts significantly promoted growth phenotypes in *B. amyloliquefaciens* in MCF-1 ($A_d \rightarrow B_r$). Higher growth indices, including cell turbidity (36–48 h), viability (36 h, $p < 0.01$), and biomass (12 h and 36 h, $p < 0.01$) were observed in the MCF-1-treated groups as compared with the respective controls (Fig. 1a–c). Furthermore, early onset of the significantly higher biofilm formation was recorded for MCF-1 treated *Bacillus* cultures (24–36 h, $p < 0.01$) compared with the controls (Fig. 1d).

*Metabolites from* B. amyloliquefaciens *suppressed* A. oryzae *growth and conidiation.* Analysis of the MCF-2 ($B_d \rightarrow A_r$) results revealed that late-log phase metabolite extracts from *B. amyloliquefaciens* culture displayed fungistatic effects on *A. oryzae* (Fig. 1e, f). Following the MCF-2, a significantly lower mycelial growth was observed for 120 h ($p < 0.01$) and 168 h incubated *A. oryzae* cultures as compared to the controls. Further, we recorded a significantly lower conidia density in MCF-2 cross-fed *A. oryzae* cultures at 168 h ($p < 0.01$) compared with the control. The inhibitory effects of *B. amyloliquefaciens* metabolites on *A. oryzae* conidiation were transient as the higher conidia density was recorded for 216 h ($p < 0.01$) incubated *A. oryzae* cultures following the MCF-2, as compared with the controls.

### MCF modulate endogenous metabolites secreted by receiver species

*Metabolites from* A. oryzae *reduced the cyclic lipopeptides (CLPs) secretion in* B. amyloliquefaciens. We examined the time-correlated exometabolomes of *B. amyloliquefaciens* subjected to MCF-1 ($A_d \rightarrow B_r$) treatment with *A. oryzae* culture extracts. Multivariate analysis (MVA) based on LC-MS/MS data sets displayed a clear disparity between the metabolite profiles of the cross-fed *Bacillus* cultures (MCF-1) and the control sets. The unsupervised principal component analysis (PCA) score plot showed an overall variability of 26.14% (PC1 = 16.50%; PC2 = 9.64%) with the data sets segregated temporally between the cross-fed and control groups during the initial growth stages (up to 12 h). However, the datasets representing later stages (24–48 h) of growth were clustered together (Fig. 2a). The supervised partial least squared–discriminant analysis (PLS-DA) score plot also highlighted temporal segregation between the MCF-1 treated and control sets across PLS1 (Fig. 2b). PLS-DA showed an overall variance of 21.25% (PLS1 = 9.55%; PLS2 = 11.90%) among the data sets and indicated 27 significantly discriminant metabolites based on their variable importance in projection (VIP) at >0.7 and $p < 0.05$. From the significantly discriminant variables between the MCF-1 treated and control sets, we characterized 17 metabolites of bacterial origin (mostly the CLPs) and three metabolites of fungal origin re-extracted from *B. amyloliquefaciens* cultures, and eight non-identified (N.I) entities (Supplementary Table 1). The PLS-DA model was evaluated with reliable goodness-of-fit parameters, including $R^2X$ (0.302), $R^2Y$ (0.987), and $Q^2$ (0.885).

Cross-fed *A. oryzae* metabolites that were re-extracted and characterized from *B. amyloliquefaciens* cultures include oxylipins 9,12,13-trihydroxyoctadec-10-enoic acid (9,12,13-TriHOME) and 12,13-dihydroxy-9-octadecenoic acid (12,13-DiHOME). In addition, sphingofungin B and a non-identified (N.I. 1) compound of fungal origin were also detected from the bacterial culture. After 12 h of incubation, we recorded a rapid depletion of the cross-fed *A. oryzae* metabolites from *B. amyloliquefaciens* cultures (Fig. 2c). Considering the endogenous metabolites of *B. amyloliquefaciens*, CLPs constituted the largest proportion of the detected compounds which mostly included the iturins, fengycins, and surfactins. Notably, the cross-fed *B. amyloliquefaciens* cultures displayed a lower relative abundance of most CLPs as compared to the control cultures (Fig. 2c). However, significantly higher levels of linear surfactins, including B-C16 (m/z 1068), A-C15

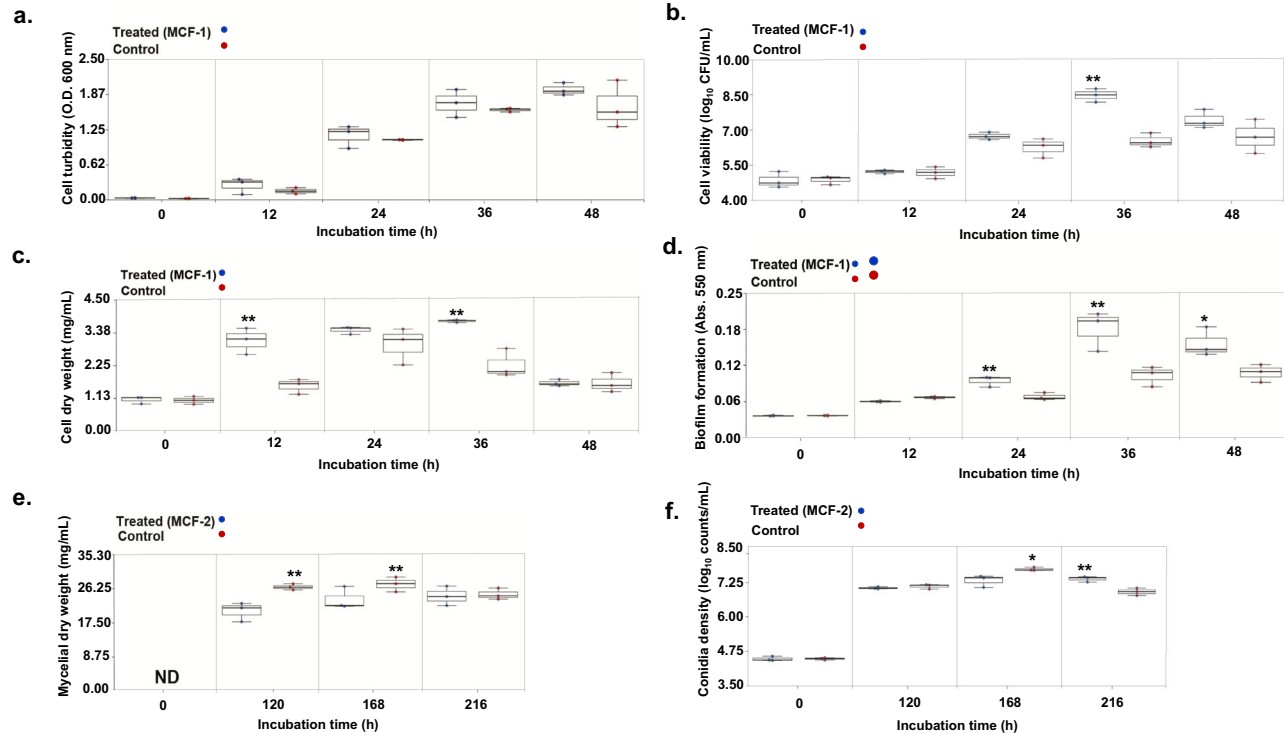

**Fig. 1 *Aspergillus* culture extracts promoted *Bacillus* growth and biofilm formation, whereas the *Bacillus* extracts suppressed mycelial growth and conidiation in *Aspergillus*.** Following the MCF-1 ($A_d \rightarrow B_r$), *B. amyloliquefaciens* phenotypes; **a** cell turbidity, **b** cell viability, **c** cell dry weight, and **d** biofilm formation was significantly enhanced in cross-fed cultures compared with the controls. In MCF-2 ($B_d \rightarrow A_r$), *A. oryzae* growth phenotypes; **e** mycelial dry weight, and **f** conidia density was significantly inhibited in cross-fed cultures compared with the controls. The statistical significance for the data was evaluated using the unpaired sample $t$ test with $p < 0.01$** and $p < 0.05$*. ND: not detected. All experiments were performed maintaining three independent biological replicates ($n = 3$). Source data for the phenotypes shown here are provided in Supplementary Data 1.

(m/z 1054), B-C15 (m/z 1040), B-C14 (m/z 1026), and B-C13 (m/z 1012) were recorded for the MCF-1 cross-fed bacterial cultures. Further, a hybrid PK-NRP compound, dihydrobacillaene, was characterized from *B. amyloliquefaciens* cultures with its lower relative abundance in cross-fed samples compared to the respective controls. Non-identified metabolites of *Bacillus* origin including N.I. 2, 3, and 4 were significantly lower in cross-fed cultures as compared to the controls. In contrast, N.I. 7 was relatively higher in cross-fed cultures, however, the remaining metabolites displayed a similar abundance among both the cross-fed and control *Bacillus* cultures.

*Metabolites from* B. amyloliquefaciens *rewired oxylipin production in* A. oryzae. The unsupervised PCA score plot indicated clear segregation between the metabolite profiles of *A. oryzae* subjected to MCF-2 ($B_d \rightarrow A_r$) and control groups, with an overall variance of 26.00% (PC1 = 16.00%; PC2 = 10.00%) (Fig. 3a). Similar patterns were evident in the PLS-DA score plot with the data sets of cross-fed samples clustered separately from the controls across PLS1 (Fig. 3b). PLS-DA showed an overall variance of 25.20% (PLS1 = 12.70%; PLS2 = 12.50%) between the cross-fed and control samples with goodness-of-fit parameters of R²X (0.306), R²Y (0.999), and Q²Y (0.931). Based on the PLS-DA model, we selected 23 significantly discriminant metabolites (VIP > 0.7, $p < 0.05$) which contributed most to the observed variance in metabolite profiles of *A. oryzae* (Supplementary Table 2). Cross-fed *A. oryzae* cultures (MCF-2) displayed 13 metabolites of *B. amyloliquefaciens* origin including 1 iturin and 12 surfactin (cyclic and linear). Notably, the cyclic surfactins including B-C16 (m/z 1050), A-C15 (m/z 1036), B-C15 (m/z 1022), B-C14 (m/z 1008), and B-C13 (m/z 994) were significantly depleted after 120 h of incubation in cross-fed *A. oryzae* cultures (Fig. 3c). However, a higher temporal abundance of

linear surfactins including B-C16 (m/z 1068), A-C15 (m/z 1054), B-C15 (m/z 1040), B-C14 (m/z 1026), and B-C13 (m/z 1012) was evident for MCF-2 treated *A. oryzae* cultures. We did not detect other CLPs (except iturin A-C15) from fungal broth upon re-extraction following the MCF-2. Iturin concentration remained roughly the same throughout incubation (except at 168 h) in cross-fed *A. oryzae* cultures. We noted a marked disparity in the endogenous metabolite levels between the cross-fed (MCF-2) and control fungal cultures. Endogenous metabolites including a polyketide (citreoisocoumarin), a sesquiterpenoid (asperaculin A), an alkaloid (sphingofungin B), and five oxylipins namely 9,12,13-TriHOME, 5,8-dihydroxyoctadeca-9,12-dienoic acid (5,8-DiHODE), 9-hydroperoxy-11,12-octadecadienoic acid (9-HpODE), 12,13-DiHOME, and 13-hydroxyoctadecadienoic acid (13-HODE) were observed significantly discriminant. Compared to the control, a significantly higher relative abundance of citreoisocoumarin and linoleate oxylipins (5,8-DiHODE and 9-HpODE) coupled with lower levels of asperculin A and oleate oxylipins (9,12,13-TriHOME and 12,13-DiHOME) were evident for the MCF-2 treated *A. oryzae* cultures. Intriguingly, we recorded the lower relative levels of both the linoleate (C18:2) and oleate (C18:1) oxylipins for the cross-fed *A. oryzae* cultures as compared to control at 216 h. Relative concentrations of sphingofungin B, 13-HODE, and two non-identified (N.I. 1 and N.I. 2) metabolites of *A. oryzae* origin observed roughly similar for both the MCF-2-treated and control cultures.

## Metabolite determinants of *B. amyloliquefaciens* and *A. oryzae* interactions

*Oxylipin 12,13-DiHOME promoted* B. amyloliquefaciens *growth and surfactin production.* To investigate how oxylipins influence *B. amyloliquefaciens* growth, we selected 12,13-DiHOME from *A.*

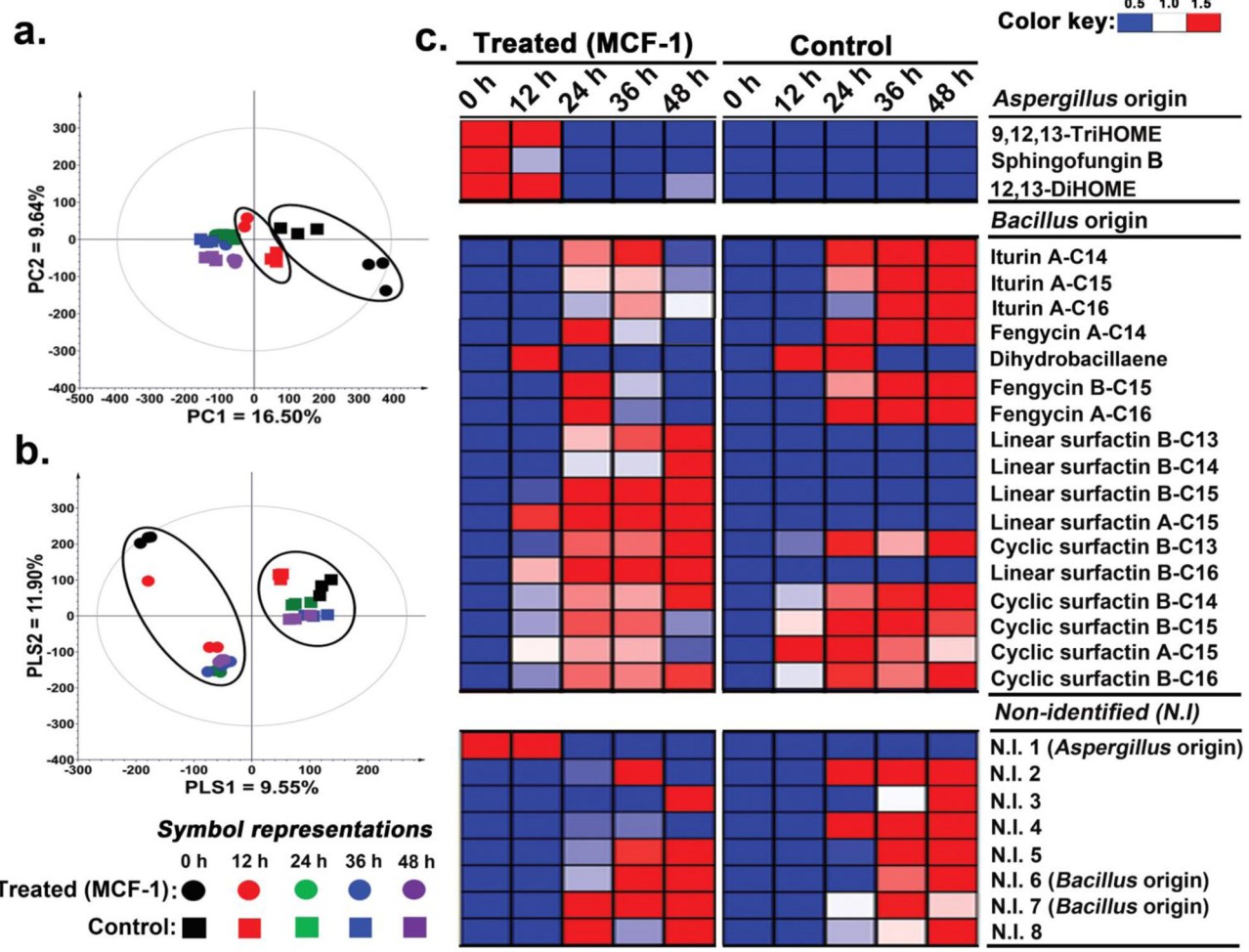

**Fig. 2 *Aspergillus* culture extracts modulated *Bacillus* metabolomes.** Time correlated **a** PCA, and **b** PLS-DA score plots for the extracellular metabolite profiles of *B. amyloliquefaciens* culture extracts based on the UHPLC-LTQ-Orbitrap-MS/MS datasets acquired in negative ion mode. **c** The corresponding heatmap based on the PLS-DA model highlighting the significantly discriminant metabolites between the *B. amyloliquefaciens* cultures subjected to MCF-1 (A$_d$→B$_r$) and control sets. Heatmap shows different headings for the cross-fed metabolites of *A. oryzae* origin, endogenous metabolites of *B. amyloliquefaciens*, and the non-identified metabolites. All experiments were performed maintaining three independent biological replicates ($n = 3$). Source data files for **a–c** are provided on sheets 1 and 2, respectively, of Supplementary Data 2.

*oryzae* extracts based on the following two criteria: (1) it was the only oleate oxylipin (other than 9,12,13-TriHOME) that was re-extracted following the MCF-1 (A$_d$→B$_r$) from *B. amyloliquefaciens* cultures, and (2) it is the only dihydroxy oleate derivative biosynthesized with fatty acid diol synthases (FADS). Reportedly, oleate oxylipins biosynthesized with FADS activity influence bacterial physiology, flagellar motility, and biofilm formation[20].

*B. amyloliquefaciens* cross-fed with standard oxylipin (12,13-DiHOME) displayed significantly higher growth indices (cell turbidity, viability, and dry weight) than the control sets (Fig. 4a–c). Unlike MCF-1, we observed a transient increase (≤12 h) in biofilm formation for the cross-fed *B. amyloliquefaciens* cultures. Biofilm formations were significantly higher in control bacterial cultures between 24 and 48 h ($p < 0.01$) as compared with those treated with standard oxylipin (Fig. 4d). Being a cyclical phenomenon, both the oxylipin treated and control cultures displayed lower biofilm formation during the later stages of incubation. Considering the metabolomes, MVA (PCA and PLS-DA) highlighted a significant variance between the oxylipin treated and control *B. amyloliquefaciens* cultures (Supplementary fig. 1a and b). Based on the S-plot of orthogonal projection to latent structures-discriminant analysis (OPLS-DA) model, we selected the endogenous metabolites as the biomarkers

which signify their high variance and correlations within the data sets representing the 12,13-DiHOME treated and control *B. amyloliquefaciens* cultures (Fig. 4e). Following the fast depletion of 12,13-DiHOME during initial growth stages, oxylipin-treated bacterial cultures showed a higher relative abundance of cyclic and linear surfactins coupled with lower levels of iturins, dihydrobacillaene, and most fengycins except A-C14 derivative (Supplementary fig. 1c).

*Cyclic surfactin A-C15 suppressed growth and metabolism in* A. oryzae. Based on the metabolite profiling of the *B. amyloliquefaciens* extracts cross-fed to *A. oryzae* cultures in MCF-2, we concluded that surfactins could be the major determinants of *Bacillus–Aspergillus* interactions. Considering cyclic surfactin A-C15 as the representative of *B. amyloliquefaciens'* CLPs re-extracted from the cross-fed *A. oryzae* cultures, we tested its effects on the ecological fitness of fungal partner.

Notably, the mycelial weights for the surfactin-treated *A. oryzae* remain temporally unvaried but significantly lower compared with the control between 120 and 216 h (Fig. 5a). Moreover, the treated sets also showed significantly lower conidia density compared with the respective controls, $p < 0.01$ (Fig. 5b). MVA (PCA and PLS-DA) highlighted a marked disparity in the

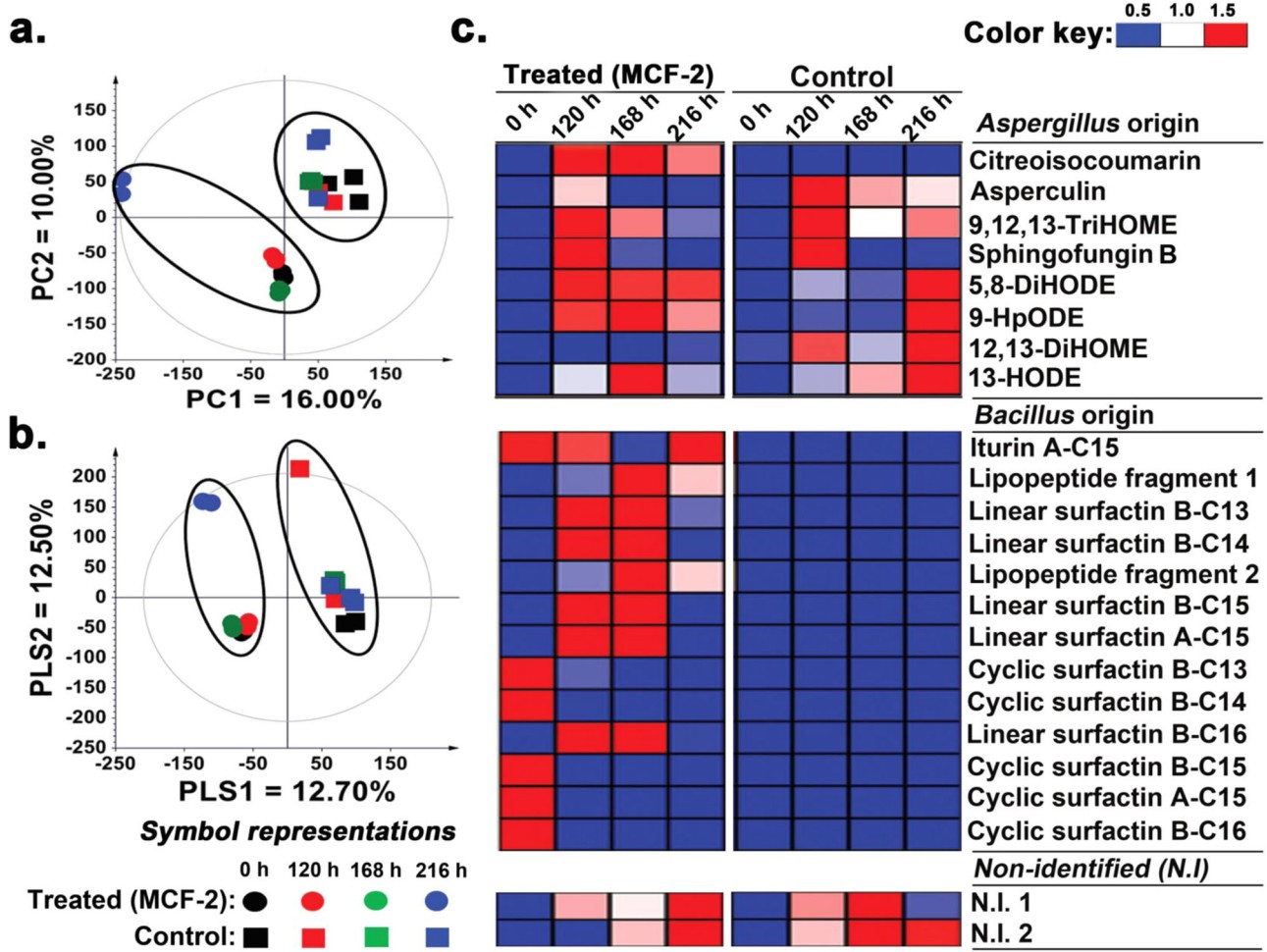

**Fig. 3 *Bacillus* culture extracts modulated *Aspergillus* metabolomes.** Time correlated **a** PCA, and **b** PLS-DA score plots for the extracellular metabolite profiles of *A. oryzae* culture extracts based on the non-targeted UHPLC-LTQ-Orbitrap-MS/MS data sets acquired in negative ion mode. **c** The corresponding heatmap based on the PLS-DA model highlighting the significantly discriminant metabolites between the *A. oryzae* cultures subjected to MCF-2 ($B_d{\rightarrow}A_r$) and control sets. Heatmap shows different headings for the cross-fed metabolites of *B. amyloliquefaciens* origin, endogenous metabolites of *A. oryzae*, and the non-identified (N.I) metabolites. All experiments were performed maintaining three independent biological replicates ($n = 3$). Source data files for **a–c** are provided on sheets 1 and 2, respectively, of Supplementary Data 3.

metabolite profiles of the surfactin treated and control *Aspergillus* cultures (Supplementary fig. 2a and b). The OPLS-DA derived S-plot highlighted the metabolite biomarkers, which demarcated the observed variance between the surfactin treated and control *A. oryzae* cultures (Fig. 5c). In corroboration with MCF-2, cyclic surfactin A-C15 (m/z, 1036) concentration was decreased while that of its linear derivative (m/z, 1054) increased temporally in treated *A. oryzae* cultures. Endogenous metabolomes for surfactin-treated *A. oryzae* cultures were characterized with a lower relative abundance of most oxylipins (5,8-DiHODE, 9,12,13-TriHOME, 12,13-DiHOME, and 13-HODE) except 9-HpODE as compared with the control. Further, we observed lower levels of citreoisocoumarin and asperculin A coupled with a significantly higher abundance of sphingofungin B for surfactin-treated *A. oryzae* as compared with control (Supplementary fig. 2c).

***Bivariate correlations deconstruct metabolite-mediated interactions.*** Pearson's correlation networks showed how the variations in the metabolomics data influence phenotypes in receiver species. Cross-fed metabolites from donor species are either consumed or transformed and hence are depleted or enriched, respectively, by the receiver species. If the depletion of the cross-

fed metabolites was concomitant with enhanced phenotypes, we assumed their positive effects on the fitness of receiver species despite a negative correlation value. However, any structural transformation of the cross-fed metabolites followed by the diminished phenotypes would correspond to have negative effects on the receiver's fitness, notwithstanding its positive statistical correlation values. In contrast, any variation in the endogenous metabolite levels would establish a direct correlation with phenotypes in the receiver species.

In MCF-1 ($A_d{\rightarrow}B_r$), *A. oryzae* derived oleate oxylipins (9,12,13-TriHOME and 12,13-DiHOME) and an alkaloid (sphingofungin) displayed weak negative correlations ($0 > r \geq -0.4$) with *B. amyloliquefaciens*' biomass and biofilm (Fig. 6a). However, strong negative ($r \leq -0.4$) correlations were evident for 12,13-DiHOME, sphingofungin, and a non-identified metabolite (b.a.n.1) of *A. oryzae* origin on *B. amyloliquefaciens*' viability. All cross-fed metabolites displayed strong negative correlations with the culture turbidity of *B. amyloliquefaciens*. Consumption-driven depletion of cross-fed metabolites of *A. oryzae*' origin promoted higher growth phenotypes in *B. amyloliquefaciens*. This was evident by the fast depletion of 12,13-DiHOME, where the strong negative correlations were concomitant with higher growth and biofilm phenotypes in *B. amyloliquefaciens* (Fig. 6b). Considering

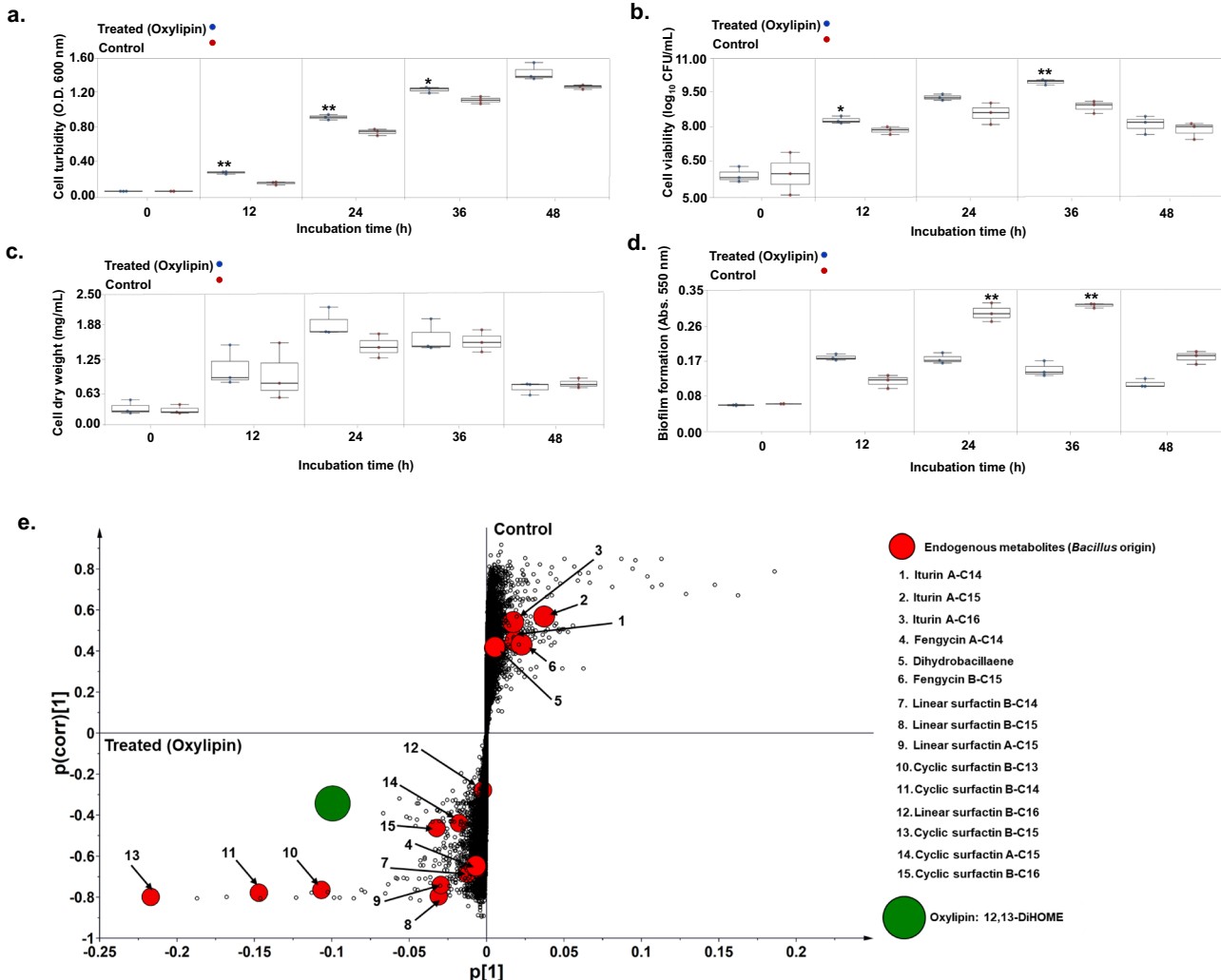

**Fig. 4 Oxylipin 12,13-DiHOME promoted higher growth and transitory biofilm formation, and enhanced surfactin secretion in *B. amyloliquefaciens*.** Effects of 12,13-DiHOME treatment on *B. amyloliquefaciens* phenotypes; **a** cell turbidity, **b** cell viability, **c** cell dry weight, and **d** biofilm formation, compared with the respective controls. Statistical significance for the data was evaluated using the unpaired sample *t* test with $p < 0.01$** and $p < 0.05$*. **e** S-plot based on the OPLS-DA model indicating the metabolite biomarkers whose levels varied significantly between the treated (oxylipin: 12,13-DiHOME) and control groups. Large & green-colored circle indicates oxylipin 12,13-DiHOME whereas small & red colored circles represent endogenous metabolites of *B. amyloliquefaciens*. The data represent the mean (±SD) of the values corresponding to three independent biological replicates ($n = 3$) used in the study. Source data files for **a**–**d** and **e** can be found on sheets 1 and 2, respectively, Supplementary Data 4.

the endogenous metabolites of bacterial origin, most CLPs which are temporally synthesized and secreted during growth displayed positive correlations ($0 < r \leq 0.4$ or $r \geq 0.4$) with *B. amyloliquefaciens'* phenotypes in MCF-1 (Fig. 6a). The only exception was dihydrobacillaene which displayed strong negative correlations with all phenotypes. Like MCF-1, treating *B. amyloliquefaciens* with 12,13-DiHOME also influenced its endogenous metabolites (CLPs) which showed positive correlations (except dihydrobacillaene) with growth (Fig. 6b).

Cross-feeding *B. amyloliquefaciens* culture extracts inhibited the mycelial growth and conidiation in *A. oryzae* following MCF-2 ($B_d{\rightarrow}A_r$). We noted strong negative correlations between the cross-fed cyclic surfactins and *A. oryzae* phenotypes (Fig. 6c). However, the linearized derivatives of cyclic surfactins displayed weak positive ($0 < r \leq 0.4$) correlations with fungal phenotypes. Hydrolytic transformation of cyclic surfactins to linear derivatives did not promote *A. oryzae* fitness significantly. Cross-feeding cyclic surfactin A-C15 standard also suggested the negative effects on *A. oryzae* fitness (Fig. 6d). However, unlike MCF-2 we observed a negative correlation between the linear surfactin (A-

C15) levels and *A. oryzae'* conidiation ($r \leq -0.4$) and mycelial growth ($0 > r \geq -0.4$). Considering endogenous metabolites, mycelial growth was positively correlated ($r \geq 0.4$) with asperculin A and all oxylipins except 12,13-DiHOME (Fig. 6c). For *A. oryzae* cross-fed with standard surfactin A-C15, perturbation in endogenous metabolite levels (except sphingofungin and 9-HpODE) showed positive correlations ($r \geq 0.4$) with growth and conidiation which verified the observations recorded in MCF-2 (Fig. 6d).

## Discussion

It is believed that auxotrophies stabilize microbial interactions and turn them obligatory under nutrient-limiting ecosystems[21]. Hence, we hypothesize that the lack of nutritional dependencies, more likely among the prototrophs colonizing a nutrient-rich environment, can result in a non-obligate and transient interaction. To test this hypothesis in the context of BFIs, we excluded the nutritional dependencies between *B. amyloliquefaciens* and *A. oryzae* partners through employing a common minimal media (CMM), which supports the growth of each individual species

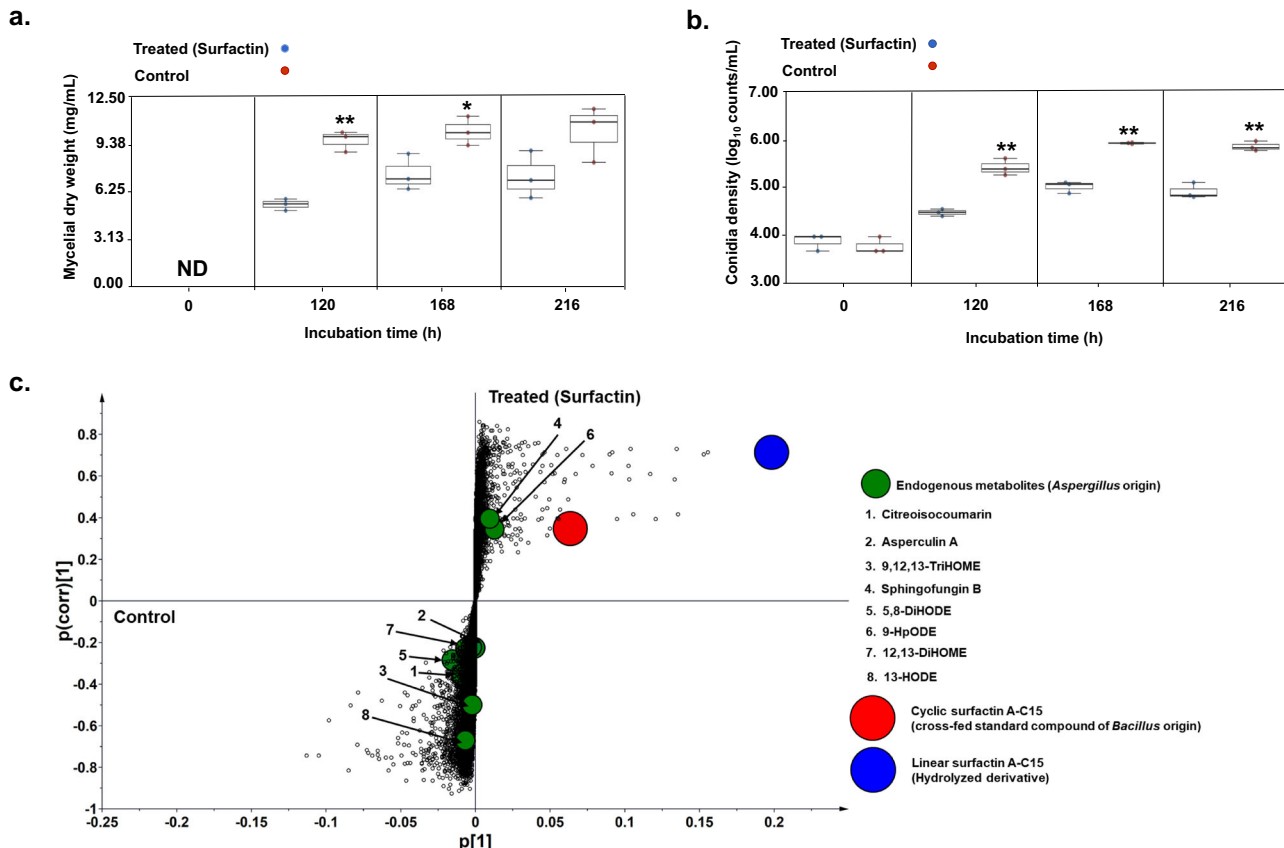

**Fig. 5 Cyclic surfactin A-C15 modulated lower mycelial growth, conidiation, and oxylipins production in *A. oryzae*.** Effects of cyclic surfactin A-C15 on *A. oryzae* phenotypes; **a** mycelial dry weight, and **b** conidia density as compared to the control sets. The statistical significance for the data was evaluated using the unpaired sample *t* test with $p < 0.01^{**}$ and $p < 0.05^*$. **c** S-plot based on the OPLS-DA model indicating the metabolite biomarkers whose levels varied significantly between the treated (surfactin) and control groups. Large circles (red: cyclic surfactin A-C15; blue: linear surfactin A-C15) indicate surfactins whereas small & green-colored circles represent endogenous metabolites of *A. oryzae*. ND: not detected. The data represent the mean (±SD) of the values corresponding to three independent biological replicates ($n = 3$) used in the study. Source data files for **a–c** can be found on sheet 1 and 2, respectively, in Supplementary Data 5.

per se but lacks the microbial compounds likely to be exchanged. Both *B. amyloliquefaciens* and *A. oryzae* were cultivated stably in CMM with their growth characteristics monitored for more than three consecutive generations. To avoid the biases arising from physical interactions between microbes, we opted to cross-feed the metabolite extracts from bacterial and fungal partners through media conditioning in pairwise MCF.

SMs are not necessarily essential for vegetative growth but are important regulators of BFIs owing to their fitness functions. SMs are mostly described to function as antibiotics and/or quorum sensing mediators[2]. Herein, we observed that late-log phase *A. oryzae* extracts promoted *B. amyloliquefaciens* growth coupled with higher and early onset of biofilm formation. Pertaining to the considerably high abundance of oxylipins in *Aspergillus* culture extracts used in cross-feeding (MCF-1), we sought to probe their potential effects on *B. amyloliquefaciens'* fitness (Supplementary fig. 3a). Oxylipins constitute an extensive class of oxygenated derivatives of polyunsaturated fatty acids which mediate intra- and inter-species signaling functions across the microbial kingdoms. Oxygenated products of oleic acid (C18:1), linoleic acid (C18:2), and linolenic acid (C18:3) are primarily involved in modulating the growth, development, nitrogen uptake, and species interactions for fungi[22,23]. Oleate oxylipins (10-HOME and 7,10-DiHOME) have also been reported to inhibit flagellum-driven motility in *Pseudomonas* which promotes biofilm formation and virulence[20]. Considering these reports, we argue that the

oleate oxylipins (9,12,13-TriHOME and 12,13-DiHOME) in *A. oryzae* extract cross-fed to *B. amyloliquefaciens* could have affected the higher and early onset of biofilm formation in MCF-1 (Fig. 1d; Supplementary fig. 3b). Linoleate-derived oxylipins, including 5,8-DiHODE, 9-HpODE, and 13-HODE, were not re-extracted from the MCF-1-treated *B. amyloliquefaciens* cultures, which can be attributed to their low stability. Heightened growth and biofilm formation in cross-fed *Bacillus* cultures were concomitant with the rapid depletion of exogenously supplied *A. oryzae* metabolites in the early log phase. Nevertheless, we continued to observe higher growth and biofilm formation in the cross-fed *B. amyloliquefaciens* cultures owing to the synergistic effects of both the characterized and uncharacterized metabolites in *A. oryzae* extracts. This points to the growth-promoting effects of *A. oryzae* metabolites, especially the oxylipins. Considering the levels of the endogenous metabolites perturbed in cross-fed *Bacillus* cultures (MCF-1), a lower relative abundance of most CLPs was evident except for the hydrolyzed linear derivatives of cyclic surfactins. Both surfactin secretion and biofilm formation are tightly regulated by quorum sensing mechanisms often involving a cascade of SMs[24]. Lower secretion of surfactin CLPs would result in proportionally higher surface tension between the bacteria and its growth surfaces which restricts swarming motility[25]. We argue that the reduced flagellar and swarming motility influenced by cross-fed oxylipins and lower surfactin secretion, respectively, could have synergistically influenced the

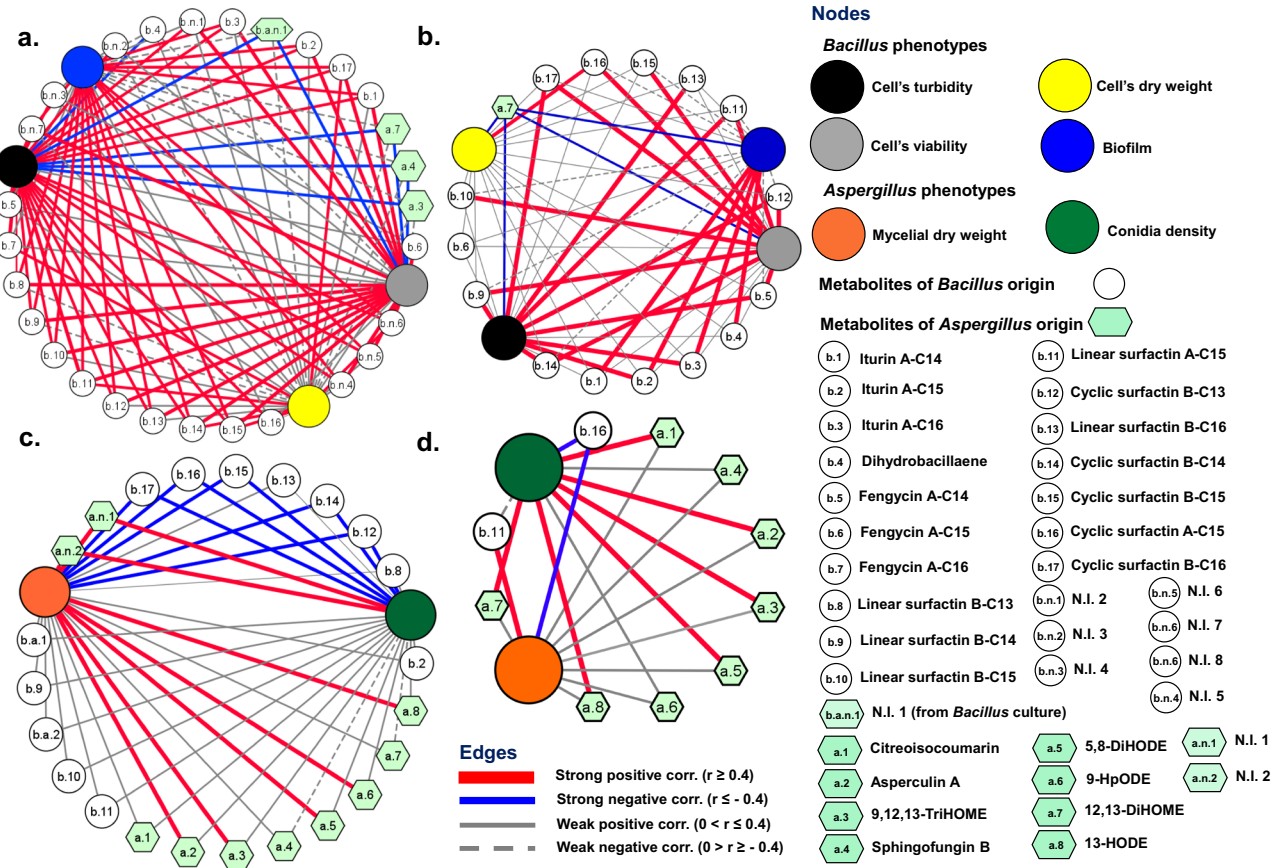

**Fig. 6 Metabolite abundance correlates with phenotypes in receiver species.** Pearson's correlation networks for **a** the cross-fed metabolites of *A. oryzae*, and **b** the standard oxylipin 12,13-DiHOME, with phenotypes of *B. amyloliquefaciens*. Conversely, correlations were visualized for **c** the cross-fed metabolites of *B. amyloliquefaciens*, and **d** the standard cyclin surfactin A-C15, with phenotypes of *A. oryzae*. In addition, we highlighted the bivariate correlations between the endogenous metabolite levels and respective phenotypes in each species. Phenotypes of *B. amyloliquefaciens* and *A. oryzae* are indicated with large & colored circular nodes. *Bacillus* metabolites are shown with small & white circular nodes, whereas *Aspergillus* metabolites are indicated with hexagon-shaped sea-green nodes. Source data for **a**–**d** can be found on sheets 1–4, respectively, in Supplementary Data 6.

higher biofilm deposition in *B. amyloliquefaciens*. Like MCF-1, *B. amyloliquefaciens* cross-fed with 12,13-DiHOME also displayed enhanced phenotypes except for the biofilm formation which suggests the transitory effects of oxylipins on bacterial motility (Fig. 4d). In addition, a significantly higher relative abundance of endogenously secreted cyclic and linear surfactins was evident for the oxylipin-treated *Bacillus* cultures. Altogether, this corroborates our previous observation that a condition lacking oleate oxylipins and/or a higher abundance of CLPs was conducive toward a low biofilm deposition. However, further studies are needed to ascertain the role of linear surfactins in *Bacillus*.

*Bacillus* CLPs are considered the key regulators of antagonistic interactions and provide the bacterial partner a competitive edge over fungi. In the present study, *Aspergillus* cultures cross-fed with *Bacillus* extracts displayed impaired mycelial growth and conidiation following MCF-2 (B$_d$→A$_r$). Late-log phase *Bacillus* culture extracts used in cross-feeding consisted of three main CLPs classes, including iturins, fengycins, and surfactins (besides some linear derivatives), and an antibiotic compound dihydrobacillaene (Supplementary fig. 4a). However, we mainly re-extracted cyclic surfactins and their linear derivatives from *Aspergillus* cultures following MCF-2 (Supplementary fig. 4b). Surfactins are composed of a heptapeptide (L-Glu[1]-L-Leu[2]-D-Leu[3]-L-Val[4]-L-Asp[5]-D-Leu[6]-L-Leu[7]) connected to the β-OH fatty acid chain (C$_{12}$–C$_{16}$) through a lactone bond, together forming a CLP[26]. Owing to their amphiphilic structure, surfactins interact readily with lipid bilayers, thereby altering membrane

permeability in fungal cells[27]. We observed a temporal decrease in the relative abundance of CLPs and a proportional increase in linear derivatives (only surfactins) in cross-fed *Aspergillus* cultures. This can be attributed to the growth-linked production of surfactin hydrolases in *Aspergillus*, as reported previously for bacterial species[28]. Intriguingly, *Aspergillus* phenotypes displayed strong negative correlations with cyclic surfactins but moderately positive or neutral correlations with linear surfactins. This suggests a loss of antifungal function, more precisely the fungistatic effects of cyclic surfactins following their structural linearization. Among the endogenous metabolites, a higher abundance of linoleate oxylipins in cross-fed *Aspergillus* cultures might be associated with the fungal resilience under challenging growth conditions. Recently, Niu et al.[29] have suggested the growth modulatory functions of linoleate oxylipin 5,8-DiHODE for regulating the lateral hyphal branching in *Aspergillus* species. However, we observed a lower abundance of oleate oxylipins in cross-fed *Aspergillus* cultures suggesting a biosynthetic rewiring of different oxylipins in cross-fed cultures. Together with linoleate (C18:2) and linoleniate (C18:3) derivatives, oleate (C18:1) oxylipins are believed to constitute precocious sexual inducers (Psi), which regulate the asexual/sexual spore formation in *Aspergillus*[22]. Hence, we suggest that a lower relative abundance of oleate oxylipins could be associated with reduced conidiation (asexual spores) in the cross-fed (MCF-2) *A. oryzae* cultures as compared to controls. Treating *A. oryzae* with standard cyclic surfactin A-C15 verified the fungistatic effects of CLPs as

**Table 1 Chemical composition of the common minimal medium (CMM).**

| Components | Concentration |
|---|---|
| NaNO$_3$ | 2 mg/mL |
| NH$_4$NO$_3$ | 8 mg/mL |
| K$_2$HPO$_4$ | 1 mg/mL |
| KH$_2$PO$_4$ | 4 mg/mL |
| Na$_2$HPO$_4$ | 14.3 mg/mL |
| MgSO$_4$ | 0.26 mg/mL |
| CaCl$_2$ | 0.01 mg/mL |
| KCl | 0.5 mg/mL |
| FeSO4 | 0.015 mg/mL |
| EDTA | 0.015 mg/mL |
| Sucrose (carbon source)/filter sterilized | 30 mg/mL |
| pH | 6.5 |

List of components used for making CMM employed for the microbial cultivation, metabolite harvests, and metabolite cross-feeding (MCF) steps of this study.

substantiated by the impaired phenotypes and oxylipin production. Previously, we have shown that both the oxylipins production and conidia density linearly decrease in *A. flavus* under challenged growth conditions[30]. Thus, we establish that cyclic surfactins that constitute a considerable portion of CLPs in *B. amyloliquefaciens* extracts effectively inhibit *Aspergillus* growth and endogenous metabolite production, and thus act as the key regulators of ammensalic interactions.

In conclusion, we posit the likely nature of metabolite-mediated ecological interactions between *B. amyloliquefaciens* and *A. oryzae* under the non-obligate pairwise MCF conditions. This study highlights the ecological implications of MCF in BFIs beyond auxotrophies while underpinning the role of SMs in microbial fitness. Oxylipins and surfactins belonging to the *A. oryzae* and *B. amyloliquefaciens*, respectively, mediate ammensalic interactions that selectively benefit bacteria and inhibit fungi. Using the non-targeted metabolomics, we delineated the impact of pairwise MCF on the metabolic plasticity of the receiver species. Correlating the metabolomic and phenotype data, we also explained how the exogenously cross-fed metabolites perturbed both the endogenous metabolites and phenotypes in receiver species. We designed a tractable cultivation medium to study the pairwise MCF between *B. amyloliquefaciens* and *A. oryzae*, however, it will be important to understand these BFIs in food fermentative niches where substrate cross-feeding and spatial factors also add to the stochasticity of the system. We believe that the experimental approach used in this study can be leveraged to design, manipulate, and understand BFIs in various microbiomes.

## Methods

**Chemicals**. HPLC-grade acetonitrile, water, methanol, ethyl acetate, dichloromethane, and hexane were procured from Fisher Scientific (Waltham, MA, USA). Standard oxylipin 12,13-DiHOME (12,13-dihydroxy-9Z, octadecenoic acid) was purchased from Caymen Chemicals (Ann Arbor, MI, USA). Cyclic surfactin C15 was purchased from Sigma-Aldrich (St. Louis, MO, USA).

**Microbial species and culture conditions**. *Bacillus amyloliquefaciens* KCCM 43033 was procured from the 'Korean Culture Center of Microorganisms' (KCCM), Seoul, Republic of Korea. Bacterial culture was maintained in BEP agar (beef extract, 3 g/L; peptone, 5 g/L). *Aspergillus oryzae* RIB 40 (KACC 44967), was provided by the 'Korean Agricultural Culture Collection' (KACC) and maintained in Wickerhams antibiotic test medium (WATM) agar[31]. We adopted a CMM capable of supporting the growth of both the *Bacillus* and *Aspergillus* partners. CMM was partially designed based on the components used in Czapek-Dox medium for fungi and the growth medium employed by Zhi et al.[32] for *Bacillus* cultivation (Table 1). We used CMM for microbial growths, culture harvest for metabolite extractions, and pairwise MCFs.

**Culture harvest and extractions for MCF**. Actively growing *Bacillus* seed culture was centrifuged, and the resulting pellet was washed twice with 1× PBS. Pellets were reconstituted in CMM to obtain a predetermined culture density (OD 600 nm~0.8; colony forming units, CFU/mL~10$^7$/mL). subsequently, 0.1% of the inoculum was transferred into 100 mL of CMM in 500 mL Erlenmeyer flasks (baffled type). The cultures were incubated at 30 °C and 200 rpm for 48 h in a shaking incubator. *Bacillus* growth parameters, including the cell turbidity (OD 600 nm), viability (CFU/mL), and biomass (CDW, mg/mL) were recorded at every 12 h interval. The OD was measured using a laboratory spectrophotometer at 600 nm and the CFU was estimated by plating the cultures on BEP agar. Microbial cultures were centrifuged (10,776 × *g* 10 min), pellets were washed twice with 1× PBS, and the CDW was measured after the overnight drying at 65 °C. Late-log phase *Bacillus* cultures were harvested, centrifuged (10,776 rpm, 10 min) and the supernatants were subjected to liquid-liquid extraction using a solvent mixture containing methanol, dichloromethane, ethyl acetate, and hexane at 1:2:3:1. Extraction was carried out in a shaking incubator (25 °C, 200 rpm) for 12 h, and the supernatant layer was decanted for drying in a vacuum concentrator (Hanil Scientific, Korea). Dried extracts were weighed and reconstituted (10,000 ppm) in 80% methanol for the LC-MS/MS analysis. Metabolites were again vacuum-dried and reconstituted in CMM prior to the MCF.

*Aspergillus* was cultured using freshly harvested conidia from WATM agar plates incubated for 216 h. Subsequently, 0.1% of the conidia suspension (~10$^7$ conidia/mL) was inoculated into CMM (100 mL) in 500 mL Erlenmeyer flasks (baffled type) and incubated at 30 °C and 200 rpm for 216 h. Cultures were harvested temporally at 0, 120, 168, and 216 h, and analyzed for conidial density (counts/mL), mycelial dry weight (mg/mL), and metabolite extraction. The growth rates of *Aspergillus* species were estimated based on conidiation and mycelial dry weight data using the method described by Singh and Lee[33]. Culture broth corresponding to the late-log growth phase was filtered using 0.2 μm bottle-top vacuum filter (Corning Inc. NY, USA). The clear broth was subjected to liquid-liquid metabolite extraction using the same procedure as described above. Dried metabolite extracts were analyzed using the LC-MS/MS prior to MCF.

**Design of experiment for MCF**. MCF between *Aspergillus* and *Bacillus* was performed by CMM conditioning prior to the respective inoculations (Fig. 7). In MCF-1 (A$_d$→B$_r$), the crude and filtered (0.2 μm) extracts from the late-log phase (168–216 h) cultures of *Aspergillus* (A$_d$: donor) were added at a concentration of 5 mg/mL into CMM prior to *Bacillus* (B$_r$: receiver) inoculation. Conversely, MCF-2 (B$_d$→A$_r$) involved the transfer of 5 mg/mL of the *Bacillus* (B$_d$: donor) late-log phase (24–36 h) extracts into CMM before *Aspergillus* (A$_r$: receiver) inoculation. Since the microbial extracts for MCF were harvested from *Aspergillus* and *Bacillus* cultivated in CMM, the control sets were added with equivalent concentration (5 mg/mL) of the CMM extracts to normalize the effects of the growth medium components in pairwise MCFs. All experiments were performed maintaining three independent biological replicates for the MCF-treated and control sets.

**Growth and developmental phenotypes**

*Bacillus*. We examined the growth profiles for the MCF-1 (A$_d$→B$_r$) and control sets of *Bacillus*. Cultures were transferred with 0.1% of the freshly prepared seed inocula (OD~0.8; CFU~10$^7$/mL) and incubated for 48 h at 30 °C and 200 rpm. Growth was evaluated at 12 h intervals for variations in culture turbidity, viability, and biomass. In addition, we examined biofilm formation as a measure of the coordinated developmental response of *Bacillus* towards MCF. The same batch of freshly grown seed culture was centrifuged, washed twice (1× PBS), and resuspended in CMM to attain the predetermined cell density prior to inoculation (0.1%) in 1.5 mL CMM in each well of the 12-well plate (Corning Inc. NY, USA). Both the treated (MCF-1) and control sets were incubated under the same conditions as with the corresponding flask cultures. Biofilm formation was examined at every 12 h alongside flask cultures with three independent biological replicates.

The biofilm quantification assay was adopted from the method described previously by O'Toole[34]. The same batch of freshly grown seed culture was centrifuged, twice-washed (1× PBS), and resuspended in CMM to attain the predetermined cell density prior to its inoculation (0.1%) in 1.5 mL CMM in each well of the 12-well plate (Corning Inc. NY, USA). Both the treated (MCF-1) and control sets were incubated under the same conditions as with the corresponding flask cultures. Biofilm formation was examined at every 12 h alongside flask cultures with three independent biological replicates. First, the medium from each culture well was gently removed without disturbing the biofilm followed by PBS (1×) washing three times. Then, 1.5 mL of crystal violet (0.1%) solution was added to each well followed by 15 min incubation at room temperature. Crystal violet was pipetted out and the culture wells were gently washed three times with distilled water. The culture plates were dried under the hot air oven (65 °C) for 3 h and 1.5 mL of acetic acid (30%) was added to dissolve the embedded CV in biofilm for the next 15 min. Solubilized CV was transferred to another microtiter plate and the absorbance was recorded at 550 nm with appropriate blanks.

*Aspergillus*. MCF-2 (B$_d$→A$_r$) cross-fed and control cultures of *Aspergillus* were inoculated with 0.1% of freshly prepared inoculum and incubated for 216 h at 30 °C and 200 rpm. The samples were harvested at 0, 120, 168, and 216 to evaluate mycelial growth and conidiation patterns. The mycelial dry weight (mg/mL) was

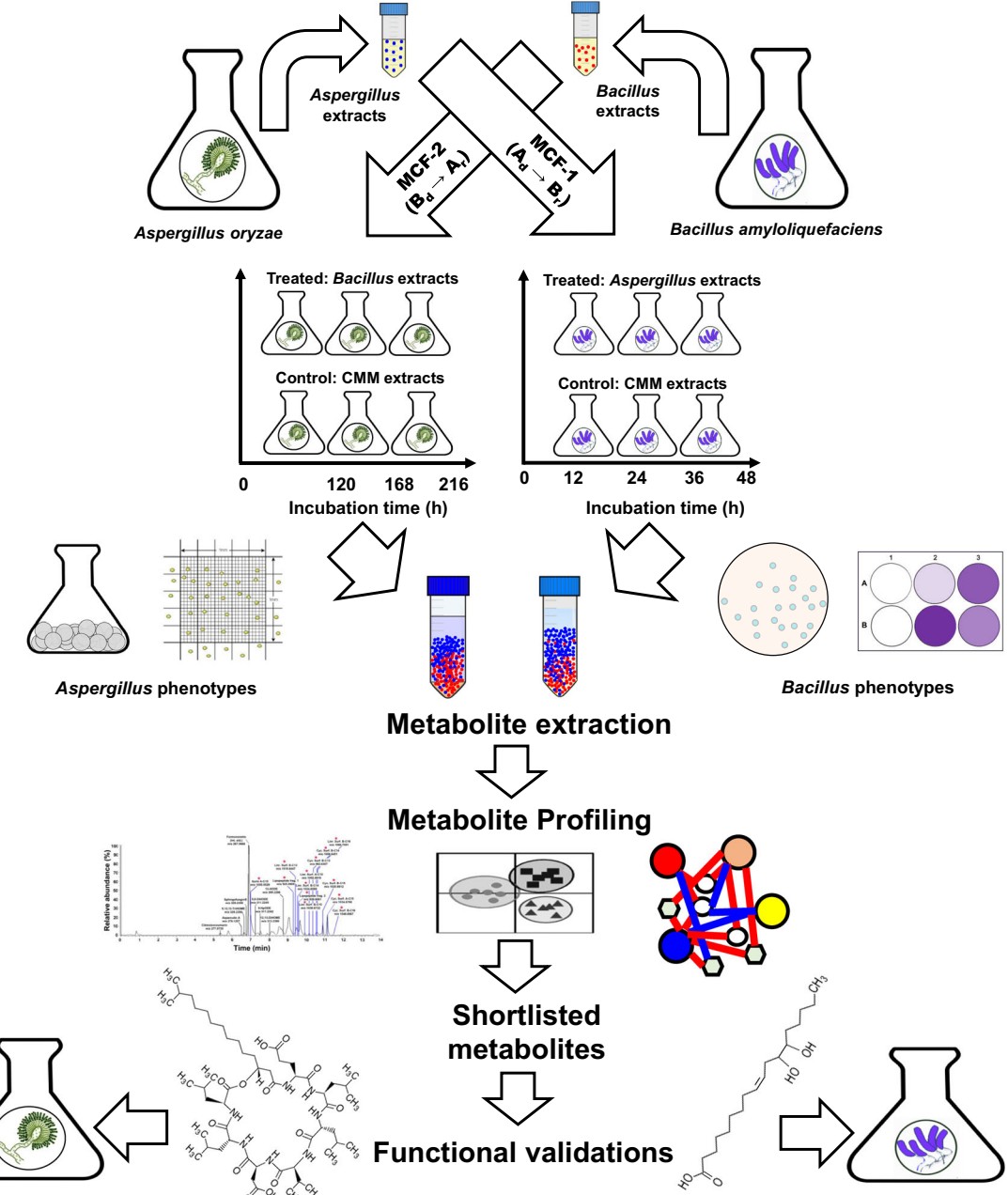

**Fig. 7 Schematics depicting the design of experiment for the non-obligate pairwise metabolite cross-feeding (MCF) between *Aspergillus oryzae* and *Bacillus amyloliquefaciens*.** Late-log phase culture extracts from *A. oryzae* as donor ($A_d$) partner were extracted and cross-fed to *B. amyloliquefaciens* as receiver ($B_r$) partner through medium conditioning in MCF-1 ($A_d{\rightarrow}B_r$). Conversely, the late-log phase culture extracts from *B. amyloliquefaciens* as donor ($B_d$) were cross-fed to *A. oryzae* as receiver ($A_r$) species in MCF-2 ($B_d{\rightarrow}A_r$). Significantly discriminant metabolites selected using non-targeted metabolite profiling were shortlisted and further validated for their potential roles in *Aspergillus-Bacillus* interactions. Bacteria and fungi cultures were harvested at different time points owing to their different growth rates, and the respective phenotypes were evaluated independently at regular intervals. All experiments were performed maintaining three independent biological replicates ($n = 3$) and appropriate experimental controls.

measured after filtering the fungal biomass through a pre-weighed filter paper (110 mm, 30 μm, Whatman Inc., Clifton, NJ, USA) and drying (65 °C) overnight. Conidial counts were examined under a compound microscope using the Neu-bauer hemocytometer. All experiments were performed maintaining three inde-pendent biological replicates.

**Non-targeted metabolite profiling**. Microbial cultures representing the cross-fed and control sets were subjected to metabolite extractions using the method described above. Dried samples were weighed and reconstituted in 80% methanol to achieve the required concentration (10,00 ppm) prior to LC-MS/MS.

Metabolite extracts were analyzed using an ultrahigh performance liquid chromatography—linear trap quadrupole—orbitrap—tandem mass spectrometry (UHPLC-LTQ-Orbitrap-MS/MS) system fitted with a vanquish binary pump (Thermo Fisher Scientific, Waltham, Massachusetts, USA). Reverse phase chromatographic separation of analytes (injection volume; 5 μL) was achieved using a C18 column (100 mm × 2.1 mm, 1.7 μm particle size), Phenomenex Kinetex, Torrance, CA, USA. The chromatographic solvent system was composed of water (A) and acetonitrile (B), each containing 0.1% formic acid (FA). The solvent gradients were run at a constant flow rate of 0.3 mL/min during the 14 min run program (5% solvent B for 1 min, 100% solvent B for 9 min, maintained for 1 min, and again 5% solvent B in last 4 min). Mass spectrometry (MS) system included LTQ-Orbitrap-Velos pro with an ion-tap (IT) and HESI-II probe. Probe heater temperature was set at 300 °C, whereas the capillary temperature and voltages were fixed at 350 °C and 2.5 kV (−ESI)/3.7 kV (+ESI), respectively. Non-targeted metabolite profiling was performed in both negative and positive ESI modes across an m/z range of 150–1500.

**Metabolite selection and functional annotations**. Based on the non-targeted metabolite profiling data for the pairwise MCF between *Bacillus* and *Aspergillus* species, significantly discriminant metabolites (VIP > 0.7, $p < 0.05$) contributing maximum toward the observed variance were selected. Cross-fed metabolites depleted maximum following the MCF was considered as the candidates most likely to have influenced the phenotypes in the receiver species (Raw data 1 and 2). We estimated the amount of significantly discriminant metabolites based on their relative peak intensities in the culture extracts used in cross-feeding. Oxylipin 12,13-DiHOME (2.40 µg/mL), representing the linoleate-derived hydroxy fatty acids of *Aspergillus* origin was added to the CMM prior to *Bacillus* inoculation. Similarly, cyclic surfactin A-C15 (5.1 µg/mL) of *Bacillus* origin was cross-fed to *Aspergillus* cultures for validations. Three independent biological replicates were maintained for each cross-fed and control set.

**Statistics and reproducibility**. Statistical significance of the phenotype data sets was examined using the unpaired sample *t* tests with PASW Statistics 18 software packages (SPSS Inc. Chicago, IL, USA).

Raw LC-MS/MS data files were converted to NetCDF format using the built-in software (Thermo Xcalibur 2.2, Waltham, MA, USA) and subjected to alignment for significant peak-picking, mass artifact filtration, baseline correction, RT shift corrections, and accurate mass calculation using MetAlign (Version 041012, RIKILT-WUR Institute of Food Safety). Peak aligned and noise subtracted data were examined for class-wise variations among the data sets using MVA in the SIMCA-P+ software (v 12.0, Umetrics, Umea, Sweden). The bivariate Pearson's correlation between metabolite abundance and corresponding phenotypes was estimated endogenous metabolomes for receiver species were estimated using PASW statistics. Correlation networks were visualized using Cytoscape software v3.7.2[35].

**Reporting summary**. Further information on research design is available in the Nature Research Reporting Summary linked to this article.

## Data availability
LC-MS/MS raw data files related to this study are available from the corresponding author upon reasonable request. Source data for microbial phenotypes, LC-MS/MS metabolite profiling, and associated statistical correlations are presented in Supplementary Data 1–6.

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

## Acknowledgements
This work was supported by the Konkuk University Researcher Fund in 2020. The work was also supported by the Korea Institute of Planning and Evaluation for Technology in Food, Agriculture, and Forestry (IPET) through the Agricultural Microbiome R&D Program, funded by the Ministry of Agriculture, Food, and Rural Affairs (MAFRA) (918011-04-1-SB010). In addition, this study was supported by the Basic Research Lab program (Grant No. 2020R1A4A1018648), through the National Research Foundation grant funded by the Ministry of Science and ICT, Republic of Korea. Further, this research was supported by the Traditional Culture Convergence Research Program through the National Research Foundation of Korea (NRF), funded by the Ministry of Science, ICT & Future Planning (NRF-2017M3C1B5019303). We thank professor Deok Kun Oh from Konkuk University, Seoul, for providing us with the purified linoleate oxylipin (5,8-DiHODE) used for the oxylipin characterization in this study.

## Author contributions
D.S. and C.H.L. contributed to the concept and design of the study. D.S. conducted the experiments and data analyses. S.H.L. helped in experiments and data analyses. D.S. wrote the manuscript with guidance from C.H.L.

## Competing interests
The authors declare no competing interests.

**Additional information**

