## [Peer Review File · Communications Biology]

Reviewers' comments:

Reviewer #1 (Remarks to the Author):

The major claims of the manuscript are that (1) this study explores an ecological implication of SMS in *Bacillus* and *Aspergillus* interaction. (2) *Aspergillus*-derived oxylipins favor *Bacillus* growth while *Bacillus*- derived cyclic surfactins impaired *A. oryzae* growth.

My major concern is about the rationale of the design in relation to the aims. The two microbial species used in this study are *Aspergillus oryzae* RIB40 (koji mold) and *Bacillus amyloliquefaciens* KCCM 43033. Bacterial-fungal interactions are very much dependent on the species involved so the authors should write the complete name in the abstract and in the introduction. Presently readers must look into the methods section to find out what *Aspergillus* and *Bacillus* species are studied.

Line 81-83. Why were these two species chosen? What is the rationale? *Aspergillus oryzae* RIB40 (koji mold) is only found in industrial factories; *B. amyloliquefaciens* is root-colonizing biocontrol bacterium. It is unlikely that these two species will ever "meet" in natural environment nor in controlled fermentations. I suggest to clarify/ precise the aim of the study.

Line 271-285. This sentence is not clear and needs to be rephrased.

Lines 444-450. These two sentences are too speculative. By in large, there are many sentences with the verbs "might or could". Bringing hypothesis to the discussion is a positive if it is well supported by previous work (citations) or experimental evidences.

Lines 478 and 828. Please define what a "reductionist approach" is in this study.

The statistical analysis seems appropriately done and the method section is providing enough information.

Reviewer #2 (Remarks to the Author):

The manuscript titled: "non-obligated pairwise metabolite cross-feeding suggests ammensalic interaction between *Bacillus* and *Aspergillus* species, is a well structured, well written manuscript and has novelty in it. The manuscript uses metabolomics to address the chemical communication between plant growth promoting rhizobacteria and fungi.

Here, the authors used first analyzed the metabolites of each species independently and then perform a crossing feeding experiment. Thereafter used machine learning tools to monitor the metabolic perturbation in the secreted metabolites and newly synthesized metabolite due to cross feeding. The use of PGPR as plant pathogen biocontrol is currently a hot topic and various metabolites such as cyclic lipopeptides (identified in the study) have been reported to play a crucial role in disease suppression. However, not much is known about how pathogens can reduce their effect and this study show that hydrolytic linearization is the method used by pathogen. I recommend publishing the manuscript as it is of interest for many researchers working on plant-microbe interaction and microbe-microbe interaction. Also, personally I am working in this kind of work.

Minor suggestions;

Figure 4e and figure 5D supposed to look like a S-shaped figure but are straight-line. To compute this SIMCA was used. This is a scaling issue. From my experience with SIMCA S-plot can only be generated with pareto/ center scaling. I recommend the authors address this.

Reviewer #3 (Remarks to the Author):

The study is well planned and the results are properly analyzed with appropriate statistical analysis. The results are appropriately presented and discussed convincingly. It is a good work. The manuscript may be accepted for publication in its current form.

Response to Editor/ Reviewer's Comments

Dear Editor,

Many greetings, please find enclosed the revised manuscript entitled '***Non-obligate pairwise metabolite cross-feeding suggests ammensalic interactions between Bacillus and Aspergillus species***' (Article number: **COMMSBIO-21-2493A**).

We are extremely thankful to the editor (**Communications Biology**) and the concerned reviewers for considering our manuscript toward revision. We are very grateful for the queries and explanations sought regarding this study, that has certainly enhanced our understanding of the topic and enabled us to improve the original version of the present manuscript. We wholeheartedly thank the concerned 'Editor and Reviewers' for their valuable time, attention, and corrective inputs.

We have now tried our best to address the reviewer/editor's queries in the revised manuscript subsuming the valuable comments and suggestions.

NOTE: The changes (deletions/ new insertions) in the revised manuscript are highlighted with the track changes in the document designated as 'Article file'.

Reviewer's comments:

Reviewer #1 (Remarks to the Author):

The major claims of the manuscript are that (1) this study explores an ecological implication of SMs in *Bacillus* and *Aspergillus* interaction. (2) *Aspergillus*-derived oxylipins favor bacillus growth while *Bacillus*-derived cyclic surfactins impaired *A. oryzae* growth.

My major concern is about the rational of the design in relation to the aims. The two microbial species used in this study are *Aspergillus oryzae* RIB40 (koji mold) and *Bacillus amyloliquefaciens* KCCM 43033. Bacterial-fungal interactions are very much dependent on the species involved so the authors should write the complete name in the abstract and in the introduction. Presently readers must look into the methods section to find out what *Aspergillus* and *Bacillus* species are studied. Line 81-83. Why were these two species chosen? What is the rational? *Aspergillus oryzae* RIB40 (koji mold) is only found in industrial factories; *B. amyloliquefaciens* is root-colonizing biocontrol bacterium. It is unlikely that these two species will ever "meet" in natural environment nor in controlled fermentations. I suggest to clarify/ precise the aim of the study.

Response: We are very thankful to the reviewer's careful appraisal of our manuscript. Here, we wish to address the reviewer's queries in two separate points: (I) Rationale & aim of the study and choice of the microorganisms, and (II) inclusion of the complete names of the microorganisms in suggested sections of the manuscript.

- I. My major concern is about the rational of the design in relation to the aims. The two microbial species used in this study are *Aspergillus oryzae* RIB40 (koji mold) and *Bacillus amyloliquefaciens* KCCM 43033. Line 81-83. Why were these two species chosen? What is the rational? *Aspergillus oryzae* RIB40 (koji mold) is only found in industrial factories; *B. amyloliquefaciens* is root-colonizing biocontrol bacterium. It is unlikely that these two species will ever "meet" in natural environment nor in controlled fermentations. I suggest to clarify/ precise the aim of the study.

Response: We do agree with the reviewer's assertion that *A. oryzae* (RIB 40) and *Bacillus amyloliquefaciens* (KCCM 43033) strains are primarily known for their different ecological occurrence as

Table 2
Microbial species identified using DGGE band sequences for examining the bacterial and fungal communities from the different steps of *doenjang* manufacturing in IP and mIP.

No.	Closest relative	NCBI accession No.	Preprocessing soy		Industrial process (IP)						Modified industrial process (mIP)									
			Soybean	Steaming	Drying	Meju fermentation			Brining		Aging		Drying	Meju fermentation			Cooling	Brining	Aging	
			0	1	2	3	4	17	22	40	51	81	141	2	3	4	5	5	37	97
Bacteria																				
B1	Lactococcus raffinolactis	HF 562962	+																	
B2	Bacillus sonorensis	JX986832.1			+	+	+	+	+	+	+	+	+							
B3	Bacillus licheniformis	JN215522.1																		
B4	Bacillus subtilis	JX993836.1			+	+	+	+	+	+	+	+	+							
B5	Bacillus velezensis ^a	-												+	+	+	+			
B6	Weissella cibaria	HF562960.1												+						
B7	Enterococcus lactis	HF562969.1																		
B8	Carnobacterium maltaromaticum	JX860593.1																		
B9	Tetragenococcus halophilus ^b	AP012046.1 b																		
B10	Leptobacterium flavescens	AB682149		+																
B11	Acidovorax delafieldii	GU195176.1		+																
B12	Bacillus seohaenensis	HE586585.1																		
B13	Vigna unguiculata chloroplast	JQ755301.1																		
Fungi																				
F1	Penicillium chrysogenum	JX480902.1																		
F2	Aspergillus flavus	JQ860302.1																		
F3	Aspergillus oryzae ^c	FN823241.1		+																
F4	Filobasidium elegans	DQ459626.1																		
F5	Cladosporium oxysporum	JQ966338.1																		
F6	Pyrenophora phaeocomes	JN940960.1																		
F7	Sporobolomyces roseus	JN937884.1																		
F8	Soybean (Glycine max)	X02623.1																		
F9	Pichia triangularis	AY227018.1																		
F10	Saccharomyces cerevisiae	JF715176.1																		
F11	Malassezia globosa	EU192364.1																		
F12	Zygosaccharomyces rouxii ^{b,c}	-																		

^a The strains are only inoculated at the *meju* fermentation process in modified industrial process (mIP).

^b The strains are only inoculated at the brining process in modified industrial process (mIP).

^c The strains were isolated from *doenjang* samples provided by CJ Cheiljedang corporations.

S. Lee et al. / Food Chemistry 221 (2017) 1578–1586

- The table highlights the co-occurrence of *A. oryzae*, *B. subtilis*, *B. sonorensis*, and *B. velezensis* across different fermentative stages of 'Doenjang' fermentation. However, the effects of microbe-derived metabolites remained elusive in microbial interactions and community succession.

Reference: Lee, Sunmin, Sarah Lee, Digar Singh, Ji Young Oh, Eun Jung Jeon, Hyung SeoK Ryu, Dong Wan Lee, Beom Seok Kim, and Choong Hwan Lee. "Comparative evaluation of microbial diversity and metabolite profiles in doenjang, a fermented soybean paste, during the two different industrial manufacturing processes." *Food chemistry* 221 (2017): 1578-1586.

- Further, both *A. oryzae* and *B. amyloliquefaciens* along with other *Aspergillus* and *Bacillus* species are reported as the **autochthonous mixed starters** of the soybean *meju* fermentation (Lee et al. 2018. *J Food Sci.* 83, 1723-1732). Many of these *Aspergillus* and *Bacillus* species also constitutes the natural microbiome of the soybeans or the starter (*nuruk*) used in traditional artisan or industrial fermentative processes.

Development of Safe and Flavor-Rich *Doenjang* (Korean Fermented Soybean Paste) Using Autochthonous Mixed Starters at the Pilot Plant Scale

Eun Jin Lee, Jiye Hyun, Yong-Ho Choi, Byung-Serk Hurh, Sang-Ho Choi, and Inhyung Lee 
Abstract: *Doenjang* (Korean fermented soybean paste) with an improved flavor and safety was prepared by the simultaneous fermentation of **autochthonous mixed starters** at the pilot plant scale. First, whole soybean *meju* was fermented by coculturing safety-verified starters *Aspergillus oryzae* MJS14 and *Bacillus amyloliquefaciens* zip6 or *Bacillus subtilis* D119C. These fermented whole soybean *meju* were aged in a brine solution after the additional inoculation of *Tetragenococcus halophilus* 7BDE22 and *Zygosaccharomyces rouxii* SMY045 to yield *doenjang*. Four *doenjang* batches prepared using a combination of mold, bacilli, lactic acid bacteria, and yeast starters were free of safety issues and had the general properties of traditional *doenjang*, a rich flavor and taste. All *doenjang* batches received a high consumer acceptability score, especially the ABsT and ABsTZ batches. This study suggests that flavor-rich *doenjang* similar to traditional *doenjang* can be manufactured safely and reproducibly in industry by mimicking the simultaneous fermentation of autochthonous mixed starters as in traditional *doenjang* fermentation.

Keywords: autochthonous mixed starters, biogenic amines, *doenjang*, pilot plant fermentation, sensory attributes

Table 1—Combinations of starter strains for *doenjang* fermentation at the pilot plant scale.

Batch	Whole soybean meju fermentation		Doenjang aging	
	Mold starter	Bacilli starter	LAB starter	Yeast starter
ABaT	A. oryzae MJS14	B. amyloliquefaciens zip6	T. halophilus 7BDE22	
ABaTZ	A. oryzae MJS14	B. amyloliquefaciens zip6	T. halophilus 7BDE22	Z. rouxii SMY045
ABsT	A. oryzae MJS14	B. subtilis D119C	T. halophilus 7BDE22	
ABsTZ	A. oryzae MJS14	B. subtilis D119C	T. halophilus 7BDE22	Z. rouxii SMY045

Reference: Lee, Eun Jin, Jiye Hyun, Yong-Ho Choi, Byung-Serk Hurh, Sang-Ho Choi, and Inhyung Lee. "Development of Safe and Flavor-Rich Doenjang (Korean Fermented Soybean Paste) Using Autochthonous Mixed Starters at the Pilot Plant Scale." *Journal of food science* 83, no. 6 (2018): 1723-1732.

- d. Further, both *A. oryzae* and *B. amyloliquefaciens* constitutes the major mold and bacterial species, respectively, in the traditional 'nuruk' or starter culture used in traditional soy food artisans. In addition, it also carries numerous mold, yeast, and lactic acid bacteria (Song et al. 2013, *J Microbiol. Biotechnol.* 23, 40-46).

J. Microbiol. Biotechnol. (2013), 23(1), 40–46
http://dx.doi.org/10.4014/jmb.1210.10001
First published online November 14, 2012
pISSN 1017-7825 eISSN 1738-8872

Analysis of Microflora Profile in Korean Traditional Nuruk

Song, Sang Hoon^{1,5}, Chunghee Lee², Sulhee Lee³, Jung Min Park⁴, Hyong-Joo Lee², Dong-Hoon Bai Song-Sik Yoon⁶, Jun Bong Choi⁷, and Young-Seo Park^{3*}

¹CJ Foods R&D, CJ Cheiljedang, Seoul 152-051, Korea

²Department of Food Engineering, Dankook University, Cheonan 330-714, Korea

³Department of Food Science and Biotechnology, Gachon University, Seongnam 461-701, Korea

⁴Korea Culture Center of Microorganisms, Korea Federation of Culture Collections, Seoul 120-091, Korea

⁵Department of Agricultural Biotechnology, Seoul National University, Seoul 151-921, Korea

⁶Division of Biological Science and Technology, Yonsei University, Wonju 220-100, Korea

⁷Graduate School of Hotel and Tourism, The University of Suwon, Hwaseong 445-743, Korea

Received: October 2, 2012 / Revised: October 15, 2012 / Accepted: October 17, 2012

Song, Sang Hoon, Chunghee Lee, Sulhee Lee, Jung Min Park, Hyong-Joo Lee, Dong-Hoon Bai, Sung-Sik Yoon, Jun Bong Choi, and Young-Seo Park. "Analysis of microflora profile in Korean traditional nuruk." *Journal of microbiology and biotechnology* 23, no. 1 (2013): 40-46.

A variety of *nuruk* were collected from various provinces in Korea, and their microflora profiles were analyzed at the species level. A total of 42 *nuruk* samples were collected and when the viable cell numbers in these *nuruk* were enumerated, the average cell numbers of bacteria, fungi, yeast, and lactic acid bacteria from all *nuruk* were 7.21, 7.91, 3.49, and 4.88 log CFU/10 g, respectively. There were no significant differences in viable cell numbers of bacteria or fungi according to regions collected. *Bacillus amyloliquefaciens* and *B. subtilis* were the predominant bacterial strains in most samples. A significant portion, 13 out of 42 *nuruk*, contained foodborne pathogens such as *B. cereus* or *Cronobacter sakazakii*. There were various species of lactic acid bacteria such as *Enterococcus faecium* and *Pediococcus pentosaceus* in *nuruk*. It was unexpectedly found that only 13 among the 42 *nuruk* samples contained *Aspergillus oryzae*, the representative saccharifying fungi in *makgeolli*, whereas a fungi *Lichtheimia corymbifera* was widely distributed in *nuruk*. It was also found that *Pichia jadinii* was the predominant yeast strain in most *nuruk*, but the representative alcohol fermentation strain, *Saccharomyces cerevisiae*, was isolated from only 18 out of the 42 *nuruk*. These results suggested that a variety of species of fungi and yeast were distributed in *nuruk* and involved in the fermentation of *makgeolli*. In this study, a total of 64 bacterial species, 39 fungal species, and 15 yeast species were identified from *nuruk*. Among these strains, 37 bacterial species, 20 fungal species, and 8 yeast species were distributed less than 0.1%.

Similarly, there are various other experimental reports which suggest the applications and co-occurrence of *A. oryzae* and *B. amyloliquefaciens* as the inocula or autochthonous microflora in fermented soy foods. Considering this BFI (Bacterial-Fungal Interaction) vital for fermentative bioprocesses, we explored the ecological implications of the secreted secondary metabolites from the mold and bacterial partners used in the study. **Hence, to exclusively examine the effects of *Aspergillus*-derived metabolites (oxylipins, polyketide, alkaloid, and sesquiterpenoid compounds) and *Bacillus*-derived metabolites (cyclic lipopeptides – fengycin, iturins, and surfactins), we conducted the cross-feeding study in a tailored growth medium.**

In the revised manuscript, we added this information in a concise way to the 'Introduction' section (Line 76-92), as shown below;

"Importance of recurring *Bacillus* and *Aspergillus* interactions cannot be understated in food matrices. Both *A. oryzae* and *B. amyloliquefaciens* are reported from the autochthonous mixed starters (*Nuruk*) used for soybean meju fermentation in traditional artisans (Song et al. 2013, Lee et al. 2018). Previously, we have shown that both *A. oryzae* and *B. amyloliquefaciens* are either used axenically or in-tandem toward the preparation of fermented soybean paste (doenjang meju) and koji (Lee et al. 2016; Gil et al. 2018; Seo et al. 2018). We have reported the concomitant high abundance of *A. oryzae* and various *Bacillus* species including *B. subtilis*, *B. sonorensis*, *B. velenzensis*, *B. seohaeanensis* throughout the fermentative stages of doenjang meju (Lee et al. 2017). Though the previously

published data suggest the likely interactions between *A. oryzae* and *B. amyloliquefaciens*, the exclusive role of microbial metabolites in nutrient rich food matrices is largely unexplored. Herein, we aim to explore the ecological implications of secretory secondary metabolites (SMs) in *B. amyloliquefaciens* KCCM 43033 and *A. oryzae* RIB 40 interactions beyond auxotrophies.”

II. Bacterial-fungal interactions are very much dependent on the species involved so the authors should write the complete name in the abstract and in the introduction.

Response: As suggested by the reviewer, we added the full names of *Aspergillus* and *Bacillus* species in the ‘Abstract’ and ‘Introduction’ sections. Moreover, we also write their full name in the ‘Results’ and ‘Discussion’ section to avoid any further confusion. We also made changes in the ‘Legends to the figures’ in main text and supplementary data for indicating the full names of the microorganisms.

However, the ‘Abstract’ section has a word limit of 150 words according to the journal guidelines (Communications Biology), hence we write the full name of the microbial species only once where they are mentioned first in the ‘Abstract’ section. We request the reviewer to understand our compulsions in this regard.

III. Line 271-285. This sentence is not clear and needs to be rephrased.

Response: In the revised manuscript, we rewrite this sentence & the whole paragraph as suggested by the reviewer (Line 288-301 of the revised manuscript).

“**Bivariate correlations deconstruct metabolite mediated interactions.** Pearson’s correlation networks showed how the variations in the metabolomics data influence phenotypes in receiver species. Cross-fed metabolites from donor species are either consumed or transformed and hence are depleted or enriched, respectively, by the receiver species. If the depletion of the cross-fed metabolites was concomitant with enhanced phenotypes, we assumed their positive effects on the fitness of receiver species despite a negative correlation value. However, any structural transformation of the cross-fed metabolites followed by the diminished phenotypes would correspond to have negative effects on the receiver’s fitness, notwithstanding its positive statistical correlation values. In contrast, any variation in the endogenous metabolite levels would establish a direct correlation with phenotypes in receiver species.”

IV. Lines 444-450. These two sentences are too speculative. By in large, there are many sentences with the verbs “might or could”. Bringing hypothesis to the discussion is a positive if it is well supported by previous work (citations) or experimental evidence.

Response: Agreeing with the reviewer suggestion, we rewrite these sentences as shown below (**Line 451-462 of the revised manuscript**).

“Together with linoleate (C18:2) and linolenate (C18:3) derivatives, oleate (C18:1) oxylipins are believed to constitute precocious sexual inducers (Psi) which regulates the asexual/sexual spore formation in *Aspergillus* (Tsitsigiannis and Keller. Trends Microbiol. 2007). Hence, we suggest that a lower relative abundance of oleate oxylipins could be associated with reduced conidiation (asexual spores) in the cross-fed (MCF-2) *A. oryzae* cultures as compared to controls. Treating *A. oryzae* with standard cyclic surfactin A-C15 verified the fungistatic effects of CLPs as substantiated by the impaired phenotypes and oxylipin production. Previously, we have shown that both the oxylipins production and conidia density linearly decrease in *A. flavus* under challenged growth conditions (Singh, Son, and Lee. *Sci Rep.* 2020).”

References:

- Tsitsigiannis, Dimitrios I., and Nancy P. Keller. "Oxylipins as developmental and host–fungal communication signals." *Trends in microbiology* 15, no. 3 (2007): 109-118.
- Singh, Digar, Su Young Son, and Choong Hwan Lee. "Critical thresholds of 1-Octen-3-ol shape inter-species *Aspergillus* interactions modulating the growth and secondary metabolism." *Scientific Reports* 10, no. 1 (2020): 1-14.

In the revised lines (451-462) and throughout the ‘Discussion’ part, we tried to substantiate our finding with previous studies as much as possible depending on the availability of the published data. More specifically, we used ‘could’ in our expressions where our observations are supported by published studies relevant to these findings. At some instance, we used the verb ‘might’ where the data we discussed is more likely a conjecture. However, we believe that proposing conjectures helps to make further hypotheses which can be tested experimentally under laboratory conditions through careful experimental designs.

V. Lines 478 and 828. Please define what a “reductionist approach” is in this study.

Response: We are thankful for the reviewer’s query. As per the definition, a ‘reductionist approach’ involves understanding a system through gaining insights of its individual components. In biology, we can consider reductionist approach as a ‘bottom-up’ method where the simplest & the most fundamental components of a system are studied first and added upward to delineate a complex system. The reductionist method of dissecting biological systems into their constituent parts has been used to study the chemical components of living organisms & bioprocesses. Though, a ‘systems approach or holistic methodology’ is more applicable to complex biological systems, the present study involved metabolite cross-feeding in a tailored growth medium avoiding direct physical interactions between ‘*B. amyloliquefaciens* – *A. oryzae*’. Herein, we focused more on the effects of selected class of metabolites (oxylipins and cyclic lipopeptides) and how these chemical entities manifests ‘*Bacillus-Aspergillus*’ Interactions. This study highlights the ecological implications of MCF in BFIs beyond auxotrophies while selectively underpinning the role of secondary metabolites in microbial fitness. Oxylipins and surfactins belonging to the *Aspergillus* and *Bacillus* species, respectively, mediate ammensalic interactions that selectively benefit bacteria and inhibit fungi. Overall, our approach was not holistic but rather the reductionist which focused on selected microbial species and how their discreet metabolite classes manifest their ecological interactions. We avoided the holistic biases of substrate cross-feeding, primary metabolites, and the spatial factors which governs the physical interactions in complex microbiomes. **However, to avoid any confusion among the readers, we decided to not use the word ‘Reductionist approach’ in the manuscript and hence deleted this from from the revised manuscript.**

NOTE: We did not elaborate this (reductionist approach) in the manuscript as we thought it can deviate the readers focus from the main theme of the study.

VI. The statistical analysis seems appropriately done and the method section is providing enough information.

Response: We are thankful for the reviewer’s critical appraisal of our manuscript’s methodology and data.

Reviewer #2 (Remarks to the Author):

I. **The manuscript titled:** "non-obligate pairwise metabolite cross-feeding suggests ammensalic interaction between *Bacillus* and *Aspergillus* species, is a well-structured, well written manuscript and has novelty in it. The manuscript uses metabolomics to address the chemical communication between plant growth promoting rhizobacteria and fungi.

Here, the authors used first analyzed the metabolites of each species independently and then reform a crossing feeding experiment. Thereafter used machine learning tools to monitor the metabolic perturbation in the secreted metabolites and newly synthesized metabolites due to cross feeding. The use of PGPR as plant pathogen biocontrol is currently a hot topic and various metabolites such as cyclic lipopeptides (identified in the study) have been reported to play a crucial role in disease suppression. However, not much is known about how pathogens can reduce their effect and this study show that hydrolytic linearization is the method used by pathogen. I recommend publishing the manuscript as it is of interest for many researchers working on plant-microbe interaction and microbe-microbe interaction. Also, personal I am working in this kind of work.

Response: We are extremely thankful for the reviewer's critical appraisal of our manuscript and its theme. We wish to add that we used *B. amyloliquefaciens* as a bacterial partner considering its origin in fermented foods (soybean koji, meju, and doenjang) along with *A. oryzae*. However, the reviewer's assessment has suggested us another important area, *i.e.*, bacteria-fungi interactions (BFIs) in agriculture involving the PGPRs. Making the best use of these suggestions, we will look for extending our research themes in those areas as well.

II. **Figure 4.e and figure 5.c supposed to look like a S-shaped figure but are straight-line. To compute this SIMCA was used. This is a scaling issue. From my experience with SIMCA S-plot can only be generated with pareto/ center scaling. I recommend the authors address this.**

Response: We welcome author's corrective suggestions and reset the scaling in Figure 4.e. and 5.c for SIMCA S-plots, as shown below;

Fig. 4

Fig. 5

Reviewer #3 (Remarks to the Author):

The study is well planned, and the results are properly analyzed with appropriate statistical analysis. The results are appropriately presented and discussed convincingly. It is a good work. The manuscript may be accepted for publication in its current form.

Response: We are extremely thankful for the reviewer's assessment and appreciation of our work in this manuscript.

Please do let us know if any further changes are required. We do hope that our revised version of the manuscript will now address the editor's as well as the reviewer's point of view. We thank you once again for your kind consideration towards the publication of our manuscript in the journal '**Communications Biology**'.

Yours Sincerely,

**The corresponding author
Manuscript number COMMSBIO-21-2493A**

REVIEWERS' COMMENTS:

Reviewer #2 (Remarks to the Author):

Please accept the manuscript. The authors have attended to my comment and corrected the figure.